# Emerging Risk Factors for Invasive Pulmonary Aspergillosis: A Narrative Review

**DOI:** 10.3390/jof11080555

**Published:** 2025-07-27

**Authors:** Ahmed Elkhapery, Mariam Fatima, Ayman O. Soubani

**Affiliations:** 1Department of Medicine, Rochester General Hospital, Rochester, NY 14621, USA; 2Department of Medicine, Wayne State University School of Medicine, Detroit, MI 48201, USA; 3Division of Pulmonary, Critical Care and Sleep Medicine, Wayne State University School of Medicine, Detroit, MI 48201, USA

**Keywords:** invasive pulmonary aspergillosis, risk factors, diagnosis, management

## Abstract

*Aspergillus* can cause a spectrum of diseases depending on the immune status and predisposing conditions. Invasive pulmonary aspergillosis (IPA) is classically seen in patients with severe immunocompromise, such as patients with hematologic malignancies, transplant recipients, and chronic corticosteroid use at high doses. Recently, IPA cases in patients without these classic risk factors, including those associated with severe respiratory viral infections, chronic obstructive pulmonary disease, liver failure, and critical illness, are being increasingly recognized. Delayed recognition and missed diagnoses contribute to increased mortality in these patient populations. Maintaining a high index of suspicion and implementation of systematic screening protocols in high-risk patients may help reduce missed or delayed diagnoses and improve patient outcomes. This review describes the pathophysiology, incidence, risk factors, outcomes, and diagnostic and treatment considerations in IPA in patients with emerging risk factors.

## 1. Introduction

*Aspergillus* is a saprophytic mold commonly found in soil and decaying vegetation [1]. They produce fastidious conidia (asexual spores) that are 2 to 3.5 microns, can be disseminated by air, and can survive in harsh environments until they germinate into hyphae under the right conditions [1]. Exposure occurs through inhalation of conidia, which can occur in the community or during hospitalization [2,3]. Estimates suggest that up to 200 conidia are inhaled every day by average people. While *A. fumigatus* species is responsible for the majority of invasive aspergillosis cases, *A. flavus*, *A. terreus*, and *A. niger* are also responsible for human infections [1].

*Aspergillus* can cause a spectrum of diseases depending on the immune status and predisposing conditions, but the focus of this review is on the most devastating and potentially lethal disease associated with *Aspergillus*: Invasive pulmonary aspergillosis (IPA). First described in a case report in 1953 in a patient with agranulocytosis [4], cases of invasive aspergillosis have increased with the increased use of chemotherapy and other immunosuppressive therapies. With the introduction of routine antifungal prophylaxis in patients with high-risk neutropenia [5], cases in patients with classic risk factors have seen a steady decline [6]. In contrast, cases in patients without these classic risk factors are being increasingly recognized [7].

While several studies have examined individual risk factors for IPA, such as COVID-19 [8], influenza [9], chronic obstructive pulmonary disease (COPD) [10], or primarily discussed the classic risk factors for IPA, this review provides a broader and more comprehensive overview of the emerging risk factors. Given that these risk factors often overlap, it is essential to consider all of them together to accurately assess the pre-test probability and determine the need for testing and empirical treatment. This review summarizes the available literature on emerging risk factors, including less common associations such as liver disease, immunomodulatory drugs, and new anti-cancer therapies such as CART cell therapy. The pathophysiology, incidence, and outcomes related to each of these risk factors are explored. Lastly, this review will discuss the diagnostic and treatment considerations of IPA associated with emerging risk factors.

## 2. Pathogenesis of Invasive Aspergillosis

In healthy individuals, inhaled conidia are cleared by mucociliary defenses. The mucus lining of the pulmonary airways contains pattern recognition receptors (PRRs), including collectins, secreted by type II cells and Clara cells, that bind to the carbohydrate-rich cell wall of *Aspergillus* conidia. This results in agglutination, enhancement of phagocytosis, and killing of the conidia by macrophages and neutrophils [11,12,13]. Mice with a knockout of mannose-binding lectin (MBL) and the surfactant proteins (the most common collectins) have increased susceptibility to IPA, showing the importance of these receptors in early recognition and activation of the innate immune response [14,15]. They also activate complement defenses against *Aspergillus* [16]. Other components in the mucus that may be involved in activation of the immune response include pentraxin PTX3 produced by phagocytes, with mice deficient in it having increased susceptibility to IPA [17].

Respiratory epithelial cells are involved in the defense against conidia not cleared by mucociliary defenses [18,19]. Epithelial cells have PRRs that initiate a proinflammatory response with the production of cytokines and chemokines such as IL-6, TNF-alpha, and IL-8 [20], recruiting neutrophils to defend against germinating conidia and *Aspergillus* hyphae [21]. Corticosteroids can blunt this inflammatory response, contributing to the increased risk of IPA in corticosteroid-treated patients [22].

Conidia bind to fibrinogen, laminin in the basement membrane, and fibronectin in the extracellular matrix and colonize the epithelium [23,24]. This is particularly important in the setting of lung injury due to any disease that results in increased exposure to fibrinogen, laminin, and fibronectin, contributing to the increased risk of IPA in these settings.

Similarly to respiratory epithelial cells, alveolar macrophages also recognize and phagocytose conidia using PRRs, generating a proinflammatory response with production of TNF-alpha, IL-1 beta, IL-6, IL-8, macrophage inflammatory protein one alpha, and monocyte chemoattractant protein 1 [25,26,27]. Polymorphisms in these receptors have been linked to increased risk of IPA in allogeneic hematopoietic stem cell transplant patients [28]. Once inside the macrophage phagolysosome, conidia are killed using oxidative and non-oxidative mechanisms [29,30]. Corticosteroid-mediated suppression of macrophage functions, or chemotherapy-induced depletion, increases susceptibility to IPA [22].

Once past the first lines of defense, *Aspergillus* conidia germinate into hyphae and produce a variety of proteolytic and degradative enzymes that support its growth, nutrient acquisition, and invasion [31]. The role of neutrophils appears to be most prominent at this stage of the infection, where neutrophils appear to attach to fungal hyphae and degranulate, resulting in fungal killing through oxidative and non-oxidative mechanisms [21]. Neutrophils also seem to inhibit further conidia germination through sequestration of iron [32].

Without adequate neutrophil defense, growing hyphae disseminate throughout the lung and other organs through the bloodstream. During angioinvasion, hyphal fragments break off into the bloodstream and are carried to other organs, where they invade the endothelium and result in significant damage with resultant thrombosis and infarction of the invaded regions, which create necrotic areas that are ripe for further fungal growth [33].

Given the involvement of multiple facets of the innate immune system in defense against IA, with mucociliary defense, respiratory epithelial cells, and alveolar macrophages responsible for clearance of conidia and activating the immune response, while neutrophils are involved in limiting hyphal growth and angioinvasion, the pathologic consequence of the disease differs based on the underlying immunodeficiency. This is well described in both patients and animal models of the disease, with disease in neutropenic patients or animal models with chemotherapy induced neutropenia characterized by thrombosis and hemorrhage from rapid angioinvasion and hyphael growth, while in non-neutropenic patients and animal models of IPA due to corticosteroids, the disease is characterized limited fungal growth and instead excessive inflammatory infiltrates and tissue necrosis [34,35,36].

## 3. Classic Risk Factors of IPA

Classic risk factors for invasive aspergillosis include prolonged neutropenia (<500 cells/mm^3^ for >10 days), chemotherapy treatment, hematologic malignancies, hemopoietic stem cell transplant (HSCT), graft vs. host disease, solid organ transplants, and prolonged and high-dose corticosteroid therapy, advanced AIDS and functional neutrophil disorders (e.g., chronic granulomatous disease) [37]. These are extensively covered in prior literature [37,38,39,40]. The rest of this review will focus on the emerging and underrecognized risk factors.

## 4. Emerging Risk Factors

### 4.1. Viral-Associated Invasive Pulmonary Aspergillosis (VAPA)

COVID-19 COVID-19-associated pulmonary Aspergillosis (CAPA) and influenza-associated pulmonary Aspergillosis (IAPA) are the two main entities in viral-associated pulmonary Aspergillosis. The CDC estimates that influenza has resulted in as many as 82 million illnesses in the 2024–2025 flu season, including between 610,000 and 1.3 million hospitalizations and between 27,000 and 130,000 deaths [41]. The coronavirus disease 2019 (COVID-19) pandemic resulted in more than one million deaths in the United States and more than six million worldwide [42]. With the introduction of widespread vaccination against COVID-19, the incidence and mortality rates have decreased substantially. Still, the infection has become endemic with occasional spikes in cases, hospitalizations, and deaths each year [42]. Invasive aspergillosis superinfection in the setting of COVID-19 or Influenza pneumonia is increasingly being recognized as a potentially lethal complication in these patients [9].

IAPA may occur early (within 24 h) of admission to the ICU [43], while CAPA seems to happen later in the ICU admission, with a median onset between 3 and 18 days from ICU admission [44,45,46]. Given the similar presentation on ICU admission, it is often not possible to differentiate clinically between IPA and severe influenza or COVID-19 pneumonia [9]. Hence, screening should happen on admission to the ICU and again if there is any concern for worsening hypoxemia, increased ventilatory support, increased respiratory secretions, pulmonary infiltrates, fever, or hemoptysis [9]. Patients with underlying immunodeficiency require a higher index of suspicion.

The current literature reports the incidence of IAPA and CAPA to be between 10% and 20% of critically ill patients with influenza or COVID-19, depending on the type of study, geographic location, diagnostic criteria used, and the rate of BAL [9]. A recent review from 2023 revealed that among 49 studies with more than 100 patients with severe COVID or Influenza who reported data for incidence of CAPA or IAPA, only 13 (27%) of the studies reported the number of patients who underwent BAL [47]. Interestingly, the studies that reported the number of patients who underwent BAL had a much higher incidence of IPA diagnosis (14% vs. 5%). Additionally, within the studies that reported the proportion undergoing BAL, studies that had a higher rate of sampling (>50% of patients) had a higher incidence of IPA diagnoses (19% vs. 9%) [47]. In a prospective study of mechanically ventilated patients with COVID-19 infection that instituted a screening protocol using BAL on admission, day 7, and in case of clinical deterioration, 27.7% of patients were diagnosed as probable CAPA [48]. Prospective observational studies have higher incidences than retrospective studies due to increased awareness and more thorough fungal work-up [49]. While increased sampling may result in more false positives, the negative impact of a missed diagnosis may be higher than over-diagnosing and overtreating *Aspergillus* colonization [47].

Several studies have suggested that lung hyperinflammation due to COVID-19 or Influenza pneumonia results in reduced epithelial barrier function and mucociliary clearance, as well as reduced macrophage and neutrophil ability to recognize and kill *Aspergillus* conidia and hyphae [50,51]. Additionally, an in vitro model using purified Influenza A particles showed the ability to increase hyphal growth [52].

IAPA and CAPA are typically seen in severely ill patients, with the need for mechanical ventilation and renal replacement therapy associated with CAPA in a recent meta-analysis from 2024 [44]. This study also indicated that pre-existing comorbidities, including chronic liver disease, hematological malignancies, chronic obstructive pulmonary disease (COPD), cerebrovascular disease, and diabetes, were risk factors for the development of CAPA. A recent meta-analysis including 1720 critically ill patients with Influenza identified prior organ transplantation, hematologic malignancy, immunocompromise, and prolonged corticosteroid use before admission to be risk factors for developing IAPA [53].

The use of immunomodulatory drugs in patients with COVID-19, such as corticosteroids [54] and the interleukin-6 receptor (IL-6R) blocker tocilizumab [55], may also result in an increased risk of IPA.

Patients with IAPA and CAPA have an increased risk of mortality as compared to patients with severe viral pneumonia [44]. A meta-analysis from 2024 showed that patients with CAPA had an all-cause mortality of 50%, which was more than double the mortality seen in patients with severe COVID-19 pneumonia but without CAPA (odds ratio 2.65 [2.04–3.45]) [44]. A meta-analysis of IAPA similarly showed a twofold increased mortality with IAPA compared to patients with severe Influenza but without IAPA [53]. Additionally, a positive serum galactomannan (GM) test is associated with increased mortality, as it may be a marker of angioinvasive disease [56]. Early initiation of antifungal treatment may be associated with lower mortality, and multiple studies advocate for this [9].

### 4.2. Critical Illness

Invasive aspergillosis is being increasingly recognized as a potentially fatal opportunistic infection in general, critically ill patients [57]. The incidence varies between studies depending on the case mix, comorbidities, and the diagnostic method. The true incidence is uncertain due to the diagnostic challenges in discriminating between colonization and true infection, the rare use of routine postmortem examination, and the absence of classic radiologic signs in non-neutropenic ICU patients [58]. A retrospective study by Meersseman et al. in 2004 found that 67 out of 1850 ICU admissions (3.6%) were diagnosed with proven or probably invasive Aspergillosis [57]. Interestingly, 68% of all patients underwent autopsy, reflecting a high autopsy rate not seen in many studies. A prospective multicenter study by Garnacho-Montero et al. published in 2005 included 1756 patients from 73 ICUs, who reported only 20 patients diagnosed with invasive aspergillosis per the clinician in charge (1.1%), five of which underwent autopsy, which confirmed the diagnosis [59].

According to a prospective observational study from 2015, including 563 patients with positive *Aspergillus* culture, as many as 50% of patients can be classified as proven (17%) or putative (36%) IPA using the AspICU criteria [60]. IPA was more common in patients with cancer and organ transplantation. Additionally, patients with IPA were more likely to have sepsis on admission to the ICU, require vasopressors, and need renal replacement therapy, compared to patients with colonization [60].

A multicenter prospective study in Italy reported that among 57 patients diagnosed with IA, corticosteroid use was the most common predisposing factor (44%), followed by autoimmune disease, COPD, and hematologic cancer (16%, 11%, and 11%, respectively) [61].

A retrospective study from 2004 showed that hematologic malignancy and COPD were the most common comorbidities in ICU admissions with IA, seen in 57% and 49% of patients with proven or probable IPA [57]. A meta-analysis from 2025 showed that diabetes was associated with increased risk of IPA in the ICU [62].

Studies of IPA in critically ill patients often report a higher mortality than observed in patients with hematologic malignancies. Meersseman et al. reported a mortality of 97% in proven IPA and 87% in probable IPA [57]. Garnacho-Montero et al. reported an 80% mortality in probable or proven IPA [59]. Tortorano et al. reported a mortality rate of 63% in probable or proven IPA [61]. Lastly, Taccone et al. reported a mortality of 79% in patients with proven IA, 67% in putative IA, and 38% in the colonized group [60]. Older age, history of bone marrow transplant, mechanical ventilation, renal replacement therapy (RRT), and higher SOFA were all independently associated with mortality.

A retrospective database study that specifically excluded patients with cancer, transplants, neutropenia and HIV/AIDS included 412 patients with aspergillosis admitted to the ICU from 2005 to 2008 reported a lower in-hospital mortality of 46% [63], suggesting that it may be the combination of critical illness, baseline disease comorbidities and invasive aspergillosis that results in the highest mortality. Figure 1 summarizes the risk factors of IPA in the ICU.

### 4.3. Liver Disease

The pathogenic mechanism of immune dysfunction in cirrhosis or advanced liver disease is not yet well understood [64]. Infection, often bacterial (most commonly Gram-negative bacilli, especially *Escherichia coli*), is a leading cause of death in patients with cirrhosis [65]. Damage to the liver’s reticulo-endothelial system through fibrosis and development of portal-systemic shunts, with loss or damage of Kupffer and sinusoidal endothelial cells, results in decreased surveillance function of the liver against organisms, including fungi, and reduced synthesis of pattern recognition receptors [66]. Leaky gut and increased bacterial translocation result in increased production of proinflammatory cytokines and upregulation of cell activation markers [66]. This constant activation can be overwhelmed in cases of acute decompensation, from infection or otherwise, and result in a state of immunodeficiency and “immune paralysis”, hallmarked by monocyte deactivation with low HLA-DR expression [67]. There are also abnormalities in neutrophil function, with reduced phagocytic function seen in cirrhosis [68,69]. These defects, as well as critical illness and exposure to corticosteroids in the treatment of acute hepatitis, may increase susceptibility to invasive *Aspergillus* infection.

Few studies report the incidence of IPA in patients with liver disease. A study from Germany instituted an intensive diagnostic protocol, with 84 critically ill patients with cirrhosis receiving a bronchoalveolar lavage weekly and as needed for worsening respiratory status [70]. BAL was tested by direct examination for fungi, fungal culture, and GM. Probable IPA was diagnosed using modified criteria based on the European Organization for the Research and Treatment of Cancer/Mycosis Study Group (EORTC/MSG) criteria, by adding liver cirrhosis as a host risk factor. This protocol resulted in 14% of patients receiving a diagnosis of “probable IPA” (12/84 patients), with a 100% mortality, compared to 65% mortality in patients without IPA. However, autopsy results were not reported/available. The increased use of BAL is likely responsible for the increased detection rate in this study [70]. Meanwhile, a retrospective study from France reported an incidence of 1.7% (17/986 patients) in patients with cirrhosis admitted to the ICU. COPD was strongly associated with IA, with an OR of 6.44 [71]. However, the BAL utilization rate was not reported. Lastly, a prospective cohort from 2017 that screened patients admitted with liver cirrhosis using serum GM twice weekly, with positive tests resulting in further workup using imaging and bronchoscopy, resulted in 1.3% of patients (2/150) being diagnosed with probable IPA [72]. This variation in the diagnosis rate is largely due to the differences in the screening and diagnostic criteria used, with the application of a rigorous screening protocol by Lahmer et al. resulting in the highest diagnosis rate among these studies [70].

In a review from 2002 to 2012, including 43 patients with cirrhosis and IA, 58% of patients used steroids for acute alcoholic hepatitis or other reasons [73]. The included articles were mainly case reports, and patients had a variety of comorbidities, including COPD, diabetes, and bacterial coinfections. A review from 2010, including 72 cases of IPA in patients with liver disease since 1973, reported that the most common host risk factor was receiving steroids, while fewer patients had COPD and/or diabetes. Interestingly, as many as 41.7% of patients had no other identified risk factor for IPA other than liver disease [74]. Lastly, in the study from Germany, RRT was the only statistically significant risk factor associated with IPA [70].

IA in patients with cirrhosis or liver disease is associated with very high mortality. Levesque et al. reported a mortality rate of 71% (12/17 patients) [71]. Lahmer et al. reported a 100% mortality in patients with IA, compared to 65% in patients without IPA [70]. A systematic review from 2021, including 11 studies, reported an overall mortality of 81.8%, with patients with acute on chronic liver failure and patients admitted to the ICU having a higher mortality (86.4% and 84%, respectively) compared to the rest of the patients [75]. A review of 72 cases of IPA in patients with liver disease since 1973 reported an overall mortality of 72.2%, with a trend towards higher survival in cases between 2000 and 2009 (58.3% compared to 87.1% between 1973 and 1999) [74]. Less than half of the patients received antifungal therapy, with the remaining only being diagnosed postmortem, reflecting a lack of awareness in this non-neutropenic population. A study of the National Inpatient Sample database from the United States identified 9515 patients with cirrhosis and invasive aspergillosis and reported an overall hospital mortality of 43.4% (2.8× higher than those without aspergillosis), with increased in-hospital complications including acute kidney injury, respiratory failure, mechanical ventilation and longer length of stay compared to patients with cirrhosis without IPA [76]. Lastly, a review from 2002 to 2012 reported a mortality of 53.5% among 43 patients with cirrhosis and IPA [73].

### 4.4. COPD

The literature estimates the incidence of IPA to be between 1.3 and 3.9% of patients hospitalized with exacerbation [77], based on a retrospective study from Spain that reported a 0.36% (53/14,618) incidence [78], a study from China that reported a 3.9% (39/992) incidence [79], and another study from China that reported 1.91% incidence (5/261) [80]. However, the true incidence in patients with COPD is difficult to estimate due to a lack of consistent definition in the literature, variation in infection surveillance, and the difficulty in differentiating between colonization and true invasive disease [81].

There is growing evidence that COPD is a significant risk factor for the development of IPA. A 2001 systematic review reported that COPD was the underlying disease in 1.3% (*n* = 26) of patients with IPA [82]. An autopsy-based study from 1992 reported that among 30 patients diagnosed postmortem with opportunistic pneumonia, corticosteroid therapy for COPD was the only identifiable risk factor in 8 patients, 6 of whom had invasive pulmonary aspergillosis on pathologic examination [83].

Admission to the ICU, presence of heart failure, recent (<3 months) antibiotic exposure, and recent (<3 months) exposure to systemic corticosteroids with the accumulated dosage equivalent to >700 mg prednisone were independent risk factors for IPA in a retrospective study from Spain [78]. A study of the National Readmissions Database in the United States reported that underlying lung cancer, hematologic cancer, or history of organ transplant were predictors of readmission with invasive aspergillosis after an initial admission for acute COPD exacerbation [84]. A case–control study from China identified systemic corticosteroids, treatment with three or more antibiotics during hospitalization, and more than 10 days of antibiotic treatment as risk factors for the development of IPA [79]. A risk predictive model identified advanced COPD, recent (within 1 month) use of broad-spectrum antibiotics, corticosteroids > 265 mg in the last 3 months, and serum albumin < 30 g/L as factors that can help predict IPA in hospitalized patients with acute COPD exacerbation [85]. A systematic review from 2008 identified systemic steroids as the most common risk factor (49 out of 65 patients), with a mean dose of 24 mg/day (range 15–65 mg/day) of prednisone equivalent [86]. Lastly, high doses of inhaled corticosteroids were also associated with increased risks in multiple case reports [87,88].

Patients with COPD and IPA have an exceedingly poor prognosis, with a systematic review by Samarakoon et al. from 2008 reporting a 91% mortality among 65 cases (43 patients with definite IPA and 22 patients with probable IA) [86]. This was despite 71% of patients being treated with antifungal therapy. A study from Spain reported a mortality of 71.7% (38/53) among patients with COPD and probable IPA [78], with 92.5% (49/53) patients being treated with antifungal therapy. Lastly, a study from China reported a mortality of 43.3% (13/30), with 87% of patients receiving antifungal therapies [79].

### 4.5. Other Conditions

#### 4.5.1. Solid Malignancies

The incidence of IPA in patients with solid tumors is not well-defined, as patients often have multiple other risk factors associated with IPA, such as underlying lung disease, critical illness, corticosteroid exposure, chemotherapy treatment, and neutropenia [89].

A retrospective study from 2018 of 101 patients with IPA and solid tumors reported that lung cancer was the most common underlying tumor (51/101, 50.5%), followed by head and neck cancer (19/101, 18.8%) [89]. The overall 12-week mortality was 31%. Interestingly, only 20% of patients presented with fever, 1% with hemoptysis, and 11% with pleuritic chest pain. CT findings were also nonspecific, with nodular lesions seen in 41% of patients, cavitary lesions in 14%, and mixed lesions in 11%. The halo and air-crescent signs were not described.

A retrospective study of 1711 patients with lung cancer reported that 45 patients (2.63%) developed IPA. Risk factors associated with developing IPA included metastatic disease, chemotherapy, and corticosteroid use [90]. IPA was associated with an inpatient mortality of 51.1%, with corticosteroid therapy and neutropenia being associated with higher mortality. The differentiation between recurrence or progression of the underlying malignancy and IPA poses a challenge in this patient population. Knowledge of the potential infection with *Aspergillus* and the appropriate diagnostic studies is critical for early diagnosis and management.

#### 4.5.2. Immunomodulatory Therapies

The programmed death (PD)-1 blockers Pembrolizomab and Nivolumab have been rarely associated with IPA in case reports [91,92,93], but the cases had multiple other risk factors, including corticosteroid use, metastatic cancer, and critical illness. A retrospective study of 740 patients with melanoma who received immune checkpoint blockers (cytotoxic T-lymphocyte antigen 4 (CTLA-4), programmed death receptor 1 (PD-1), or programmed death ligand 1 (PD-L1) blockers) reported serious infections in 54 patients (7.3%), but only 2 cases of IPA [94].

TNFα Blockade Therapy has been associated with invasive fungal infections, including histoplasmosis, candidiasis, and aspergillosis, with a review from 2008 identifying 64 reported cases of IPA in patients receiving TNFα blockers, with an overall mortality rate of 82% [95]. All but one patient were exposed to other immunosuppressants, limiting the ability to draw a causal link. A meta-analysis from 2008 reviewed the risk of serious infections associated with the use of rituximab, abatacept and anakinra in treatment of rheumatoid arthritis, and reported no overall increased risk of infection with rituximab or abatacept, but increased risk of infections with high doses of anakinra (≥100 mg daily), with invasive fungal infections being rare [96].

#### 4.5.3. CART Cell Therapy

Chimeric antigen receptor-modified T-cell (CAR-T-cell) therapy is a new treatment modality involving genetically modifying T lymphocytes to target certain antigens expressed by malignant cells [97]. Anti-CD19 CAR-T cell therapy was approved for B-cell malignancies in 2017 [98]. The use of CAR-T-cell therapy is currently being studied to target various antigens associated with other types of cancers. Invasive fungal infections after CAR-T-cell therapy have been reported, but the risk is not well defined due to the presence of other risk factors such as hematologic malignancies and hematopoietic cell transplant [99]. In a clinical trial of CAR-T-cell therapy for relapsed/refractory B-cell acute lymphoblastic leukemia, three patients (5.7%) developed IPA [100]. In another study of 133 patients with hematologic malignancies, only one patient (0.75%) was diagnosed with IPA [101]. Lastly, a 2022 study including 280 patients with non-Hodgkin lymphoma reported two cases (0.71%) of IPA after receiving CAR-T-cell therapy [102]. However, it is important to consider that many of these patients were receiving antifungal prophylaxis, and further research is needed to clarify the risk of IPA in CAR-T cell recipients with and without prophylaxis [103].

## 5. Clinical Features

The diagnosis of invasive aspergillosis requires a high index of suspicion and an awareness of the various risk factors that increase the pretest probability of the diagnosis. Implementation of routine screening with BAL cultures, BAL GM, and *Aspergillus* PCR in patients with respiratory failure, as well as the use of serum tests such as serum GM, can help increase the case detection rate, at the risk of increasing false positives [49,104].

Radiologic findings in neutropenic patients include the classic halo sign (a pulmonary nodule or mass surrounded by a halo of ground-glass attenuation) [58], which may suggest early angioinvasive disease [105]. However, these findings are less common in nonneutropenic patients [58]. Additionally, in patients with viral and/or bacterial pneumonias, the presence of infiltrates due to pneumonia is likely to hide any *Aspergillus*-related findings. On imaging, cavitations, solid nodules, persistent infiltrates, tree in bud signs, or bronchial wall thickening should prompt bronchoscopy with BAL in the appropriate clinical setting [106] (Figure 2). Additionally, the diagnosis should be considered in patients who are worsening or have a recurrence of fever despite antibiotics [107].

## 6. Diagnosis

Regardless of host risk factors, the diagnosis of proven IPA requires histopathologic, cytopathologic, or direct microscopic examination of a specimen from a sterile site showing fungal invasion associated with tissue damage, or isolation of *Aspergillus* by culture from a normally sterile site along with clinical or radiologic evidence of an infectious disease process [108]. Invasive diagnostic testing with biopsy is often not feasible in clinical practice due to critical illness or other comorbidities, such as thrombocytopenia, that increase the risks of the procedure [109]. Thus, the diagnosis is based on compatible clinical symptoms, radiological findings, and mycological evidence from serum or bronchoalveolar lavage testing, with the biggest point of debate being how to differentiate aspergillus colonization of the respiratory tract from true infection [110]. Vanderbeke et al. showed that the antemortem diagnosis of viral-associated or COVID-19-associated probable IPA was confirmed on postmortem autopsy in 52% of the cohort, despite the use of antifungal therapy, which potentially may have resulted in false-negative autopsy results [111]. This suggests that a high percentage of patients with “probable” IPA are likely true infection, and that focus should be shifted from confirming true infection in these patients to prompt initiation of antifungal therapy [112].

In patients with classic risk factors, the diagnostic criteria set forth by the EORTC/MSG are most commonly used [108]. The diagnosis of “Probable” IPA requires the presence of at least one host factor, a clinical feature, and mycological evidence. Patients with host factors and clinical features without mycological evidence are considered “possible” IPA. In the latest revision and update of the criteria from 2020 [108], galactomannan and *Aspergillus* PCR were added as mycologic criteria, given multiple studies showing effectiveness in confirming or excluding the disease [113].

In the absence of classic risk factors of IA, the EORTC/MSG criteria have been adapted for patients with COVID-19 infection [114], Influenza infection [115], critical illness [116], and cirrhosis [70], by adding the temporal relation to the specific condition as a host risk factor and otherwise using similar clinical features criteria and mycological evidence criteria used by the EORTC/MSG. This is performed with the caveat that many of the mycological tests were studied in hematological cancer patients [114,117]. The sensitivity and specificity of the various tests are discussed in the following section.

The Invasive Fungal Diseases in Adult Patients in Intensive Care Unit (FUNDICU) 2024 consensus provides the most recently developed definition for IPA in non-cancer patients [118]. For the diagnosis of probably IPA or ITBA, patients should have at least one of the compatible signs and symptoms, which include fever, dyspnea and worsening respiratory status despite treatment, along with at least one risk factor, which include Influenza, COVID-19, moderate/severe COPD, cirrhosis, uncontrolled HIV or solid tumors, and at least one clinical criterion and one mycological criterion. Figure 3 provides an algorithm for the diagnosis of IPA based on the FUNDICU criteria.

### 6.1. Test Performance Considerations

The sensitivity and specificity of serum and BAL fluid GM antigen or *Aspergillus* PCR have been studied almost exclusively in patients with classic risk factors such as hematologic malignancies and HSCT recipients.

A Cochrane review from 2015 reviewed the sensitivity and specificity for GM serum levels included 54 studies with 586 patients with proven or probable invasive aspergillosis, reported an overall sensitivity of 72% (95% CI 65–80%) and a specificity of 88% (95% CI 84–92%) when using a cut-off value of 1.0 optical density index (ODI), and a sensitivity of 82% (95% CI 73% to 90%) and specificity of 81% (95% CI 72% to 90%) when using a cut-off value of 0.5 ODI. However, it is notable that all the studies were performed on patients with neutropenia or functional neutropenia [119]. There are no large-scale studies in patients without neutropenia, but small studies suggest that the sensitivity for serum GM is lower in patients without the classic risk factors. A study from 2017 reported a sensitivity of 37.84% and 24.32%, and a specificity of 87.14% and 95.71%, at ODI cut-off values of ≥0.5 and ≥1.0, respectively, in patients without neutropenia [120]. It is hypothesized that this is due to the lower prevalence of angioinvasive disease in patients without neutropenia [121].

A meta-analysis from 2012 reported the sensitivity and specificity of BALF GM to be 87% and 89% at a cut-off value of 0.5 ODI, and 86% and 95% at a cut-off value of 1.0 ODI, respectively [122]. Similarly to studies on GM serum levels, all the studies exclusively recruited patients with classic risk factors for IPA. In patients without neutropenia, BALF GM has a lower sensitivity and specificity of 75.68% and 80.72% at a cut-off value of 0.5 ODI, and 64.86% and 90.36% at a cut-off value of 1.0 ODI, respectively [120]. The most recent guidelines from EORTC/MSG recommend a cut-off value of ≥1.0 ODI in a sample of blood or serum, or ≥1.0 ODI in a sample of BALF, or both blood ≥ 0.7 ODI and BALF ≥ 0.8 ODI [108]. The more recent recommendations by the FUNDICU panel suggest serum galactomannan > 0.5 ODI and BALF galactomannan ≥ 1.0 ODI [118].

A systematic review from 2009, including 16 studies, reported a sensitivity of 75% (95% CI 54–88%) and specificity of 87% (95% CI 78–93%) for two consecutive positive blood, serum, or plasma samples for *Aspergillus* PCR [123]. Once again, most patients included in these studies had classic risk factors for IPA. Although it is not certain that similar accuracy would be seen in patients without neutropenia, small studies suggest they may be comparable [124].

The role of Beta-D-Glucan is still evolving and may be helpful as part of screening or surveillance in populations at risk for invasive fungal diseases. However, it is not currently used as a diagnostic criterion in defining cases as it is not specific to aspergillosis and can be positive in infections due to *Candida* and *Pneumocystis jirovecii* [125]. It can also be positive due to a variety of conditions and treatments that are common in critically ill patients, such as hemodialysis with cellulose membranes, exposure to glucan-containing gauze, intravenous immunoglobulin, albumin, and use of certain antibiotics [126]. Another disadvantage is the lack of a standardized serum assay cut-off point for optimum performance. Hachem et al. [127], Koo et al. [128], and Persat et al. [129] reported a sensitivity of 67%, 75%, and 69%, respectively, using a cut-off of 80 pg/mL. Meanwhile, Obayashi et al. reported a 100% sensitivity using a cut-off of 30 pg/mL [130]. Lastly, these studies were performed predominantly in patients with hematologic or solid malignancies, further limiting the generalizability of the results. A study of immunocompromised ICU patients showed a sensitivity of 85.7% but a specificity of 36.4% using a cut-off of 80 pg/mL, suggesting it may be helpful as a test to rule out the infection in populations at risk [131]. Further studies are needed to define the role of Beta-D-Glucan in the diagnosis of IPA.

### 6.2. Management Considerations for Emerging Risk Factors of IPA

The diagnostic criteria described above help define cases for research purposes. In clinical practice, treatment should be considered and started promptly if there is a high index of suspicion and the clinical picture is consistent with invasive aspergillosis, without necessarily fulfilling all the criteria [132,133].

Triazole antifungal therapy, which inhibits the synthesis of the fungal membrane component ergosterol, is the preferred therapy for treating invasive aspergillosis [132]. Azoles generally have better tolerability and outcomes than liposomal Amphotericin B (L-amB) [134]. The Infectious Disease Society of America (IDSA) guidelines recommend voriconazole as the first-line therapy, with isavuconazole or L-amB as alternatives if voriconazole is unavailable or cannot be used [133]. Similarly, the European Conference on Infections in Leukemia (ECIL) and the European Society of Clinical Microbiology and Infectious Diseases (ESCMID) recommend either voriconazole or isavuconazole as first-line therapy, with L-amB as an alternative if they are not available or cannot be used [135,136].

In patients with refractory or progressive disease despite initial antifungal therapy, an individualized approach is recommended. Available options include switching the antifungal class (L-amB or Caspofungin if the initial antifungal was an azole), tapering immunosuppression, or surgery (e.g., debridement for localized disease or necrotic lesions). Additionally, combination antifungal therapy can be considered in limited situations [132,133].

The side effects of the selected therapy should be considered in the context of the patients’ comorbidities, which is particularly important in patients with critical illness. Voriconazole is hepatotoxic, and treatment with L-amB or caspofungin may be preferable in patients with acute liver failure or cirrhosis [73]. This limitation may contribute to higher mortality in selected groups of patients, such as the higher mortality seen in patients with cirrhosis. This was shown in a study from Germany that reported a 100% mortality in the 12 patients diagnosed with probable IA, with 11/12 patients being treated with L-amB, and only one patient receiving voriconazole [70]. Many drugs used in the treatment of hematologic malignancies, as well as immunosuppressants (e.g., cyclosporine, tacrolimus, and sirolimus), have significant drug–drug interactions with azoles, which are inhibitors of cytochrome P450-3A4 [132,137]. Therapeutic drug monitoring (TDM) can minimize toxicities and ensure efficacy of azoles in these cases [133].

### 6.3. Duration of Therapy

The literature does not clearly define the duration of therapy. The IDSA recommends a minimum duration of 6–12 weeks, depending on the degree of immunosuppression, site of disease involvement, and evidence of improvement [133]. Trending serum GM antigen levels are promising and may have prognostic utility, with increasing levels associated with progressive disease [138]. Further research is needed to define the optimal duration of therapy.

### 6.4. Limitations

The key literature was reviewed and summarized to highlight the current state of knowledge on this topic. However, this review is subject to inherent limitations due to its narrative nature, including selection and confirmation bias, resulting from the lack of a systematic search methodology.

### 6.5. Conclusion and Future Directions

The risk factors for IPA are expanding and are expected to become more relevant with the increase in respiratory viral infections and advances in therapeutics for a variety of malignant and non-malignant conditions. The risk factors described often co-exist in patients, especially in the ICU setting. Clinicians must consider all possible risk factors present and maintain a high index of suspicion in patients who have atypical presentations or are worsening despite initial treatment. Delayed recognition and missed diagnoses contribute to increased mortality in these patients. While implementation of systematic screening protocols may increase false positives and identification of colonization, it would reduce the risk of harm to patients due to missed and delayed diagnoses, given the high mortality associated with IPA. Early initiation of antifungal therapy is vital for improving outcomes.

## Figures and Tables

**Figure 1 jof-11-00555-f001:**
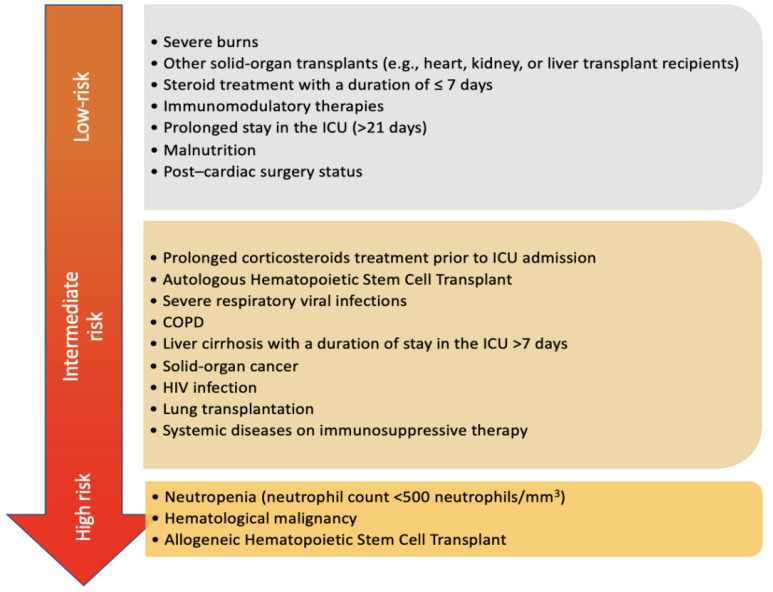
Risk factors for invasive aspergillosis in the ICU. The figure stratifies the risk factors into high, intermediate, and low risk. Neutropenia, hematologic malignancy, and allogeneic hematopoietic stem cell transplant (HSCT) confer the highest risk for invasive aspergillosis in the ICU. The figure also provides an insight into the relative risk of emerging risk factors for IPA.

**Figure 2 jof-11-00555-f002:**
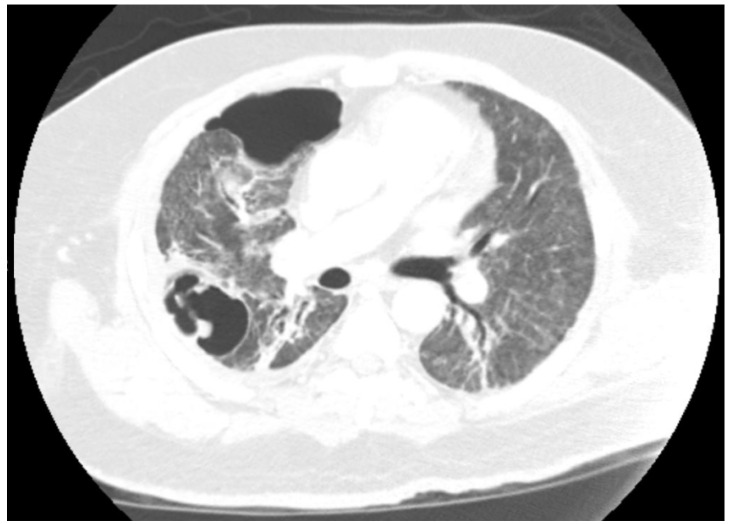
Chest CT image of a patient with COVID-19 pneumonia who required prolonged mechanical ventilation. His chest CT revealed fibrotic changes with right cavitary lesions. Bronchoscopy with bronchoalveolar lavage resulted in positive cultures for Aspergillus fumigatus, and galactomannan was elevated to 7.6 ODI. The patient was diagnosed with IPA and was started on voriconazole. He was eventually extubated and discharged home.

**Figure 3 jof-11-00555-f003:**
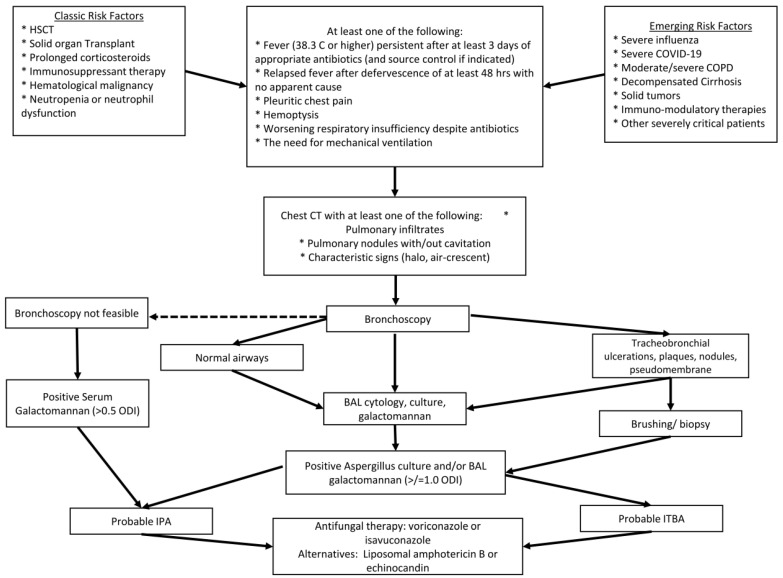
An algorithm for the diagnosis and management of IPA in critically ill patients. HSCT: Hematopoietic Stem Cell Transplant. COVID-19: Coronavirus Disease of 2019. COPD: Chronic Obstructive Pulmonary Disease. CT: Computed Tomography. BAL: Broncho-Alveolar Lavage. ODI: Optical Density Index. IPA: Invasive Pulmonary Aspergillosis. ITBA: Invasive Tracheobronchial Aspergillosis.

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
