# Peer review of "Emerging Risk Factors for Invasive Pulmonary Aspergillosis: A Narrative Review"

_jof, 2025, doi:10.3390/jof11080555_

Round 1

Reviewer 1 Report

The manuscript describes emerging risk factors of invasive pulmonary aspergillosis. It is focused on clinical insight, risk factor analyses, and considerations on managing specific populations. The manuscript structure is proper and well-designed.

There are several minor issues like lack of Latin names or words italicization, extra spaces, and others that should be corrected by authors or the journal editorial team.

 In terms of current existing knowledge, the manuscript provides an interesting literature review based on 122 references. However, only 27% of the references were published within the last five years. The manuscript would profit if the cited literature contained older and newer publications.

In my opinion, the section describing diagnostic procedures, especially on D-glucan, and treatment should be extended, and include a description of drug resistance occurring in Aspergillus species.

Lines 14-16 and 36-37 stating what is the article is about and what is describes should be similar.

Several statements should be followed by references, see lines: 106, 108, 110, 111 were you left text (REFERENCES), 113, 118, 124, 131, 153, 163, 164, 167, 174, 

Insted of placing the reference number at the ned of a sentence put it directlu after the place in the text it's refered to, e.g. line 193; Taccone et al [ref].

Line 208; "ofeten bacterial" - are there any bacterial species in particular

Line 225; explain EORTC MSG

Line 236; please specify the "diferences" nad "diagnostic criteria".

Line 245; What does RRT mean?

Section 4.4. The literature review in this section seems not to be based on recent literature; you cite several papers from 2010 and 2012, which is fine, but I would recommend confronting the data from those references with recent papers. Such a comparison of data will definitely strengthen your paper.

Lines 436-440, the role of beta-D-glucan should be elaborated. What are the pros and cons of this marker? Why is it not commonly used in clinical practice like GM?

Author Response

Reviewer 1

Major comments

The manuscript describes emerging risk factors of invasive pulmonary aspergillosis. It is focused on clinical insight, risk factor analyses, and considerations on managing specific populations. The manuscript structure is proper and well-designed.

Response: Thank you for this positive feedback

There are several minor issues like lack of Latin names or words italicization, extra spaces, and others that should be corrected by authors or the journal editorial team.

Response:  Thank you for this feedback, we made these changes and hope the editorial team can catch anything we missed.

In terms of current existing knowledge, the manuscript provides an interesting literature review based on 122 references. However, only 27% of the references were published within the last five years. The manuscript would profit if the cited literature contained older and newer publications.

Response: Thank you for this feedback. We reviewed the pubmed results for “invasive aspergillosis” and “critical illness” or “COPD” or “ICU” or “COVID” in the last 5 years, and the relevant results were incorporated into this revised manuscript where possible.

In my opinion, the section describing diagnostic procedures, especially on D-glucan, and treatment should be extended, and include a description of drug resistance occurring in Aspergillus species.

Response: Thank you for this feedback. The section was expanded to include some of the suggested information including B-D-glucan. Please note that details of treatment and drug resistance are beyond the scope of this review, as the primary focus is on the spectrum of emerging risk factors for IPA.

Detailed comments

Lines 14-16 and 36-37 stating what is the article is about and what is describes should be similar.

Response: Thank you, this was corrected

Several statements should be followed by references, see lines: 106, 108, 110, 111 were you left text (REFERENCES), 113, 118, 124, 131, 153, 163, 164, 167, 174,

Response: Thank you, this was corrected. We apologize for the oversight.

Instead of placing the reference number at the end of a sentence, put it directly after the place in the text it's referred to, e.g. line 193; Taccone et al [ref].

Response: Placing the reference at the end of the sentence is the standard per most journals. For consistency, we kept it this way for the entire manuscript. If you and the editorial staff still believe it is necessary, we can change this.

Line 208; "often bacterial" - are there any bacterial species in particular

Response: This information was added (gram negative, most common E coli)

Line 225; explain EORTC MSG

Response: Thank you, this was fixed (stands for European Organization for the Research and Treatment of Cancer/Mycosis Study Group).

Line 236; please specify the "differences" and "diagnostic criteria".

Response: The differences were explained in the paragraph- the german study by Lahmer et al et al used a screening algorithm with the use of BAL, while the other studies either did not have a screening algorithm (the French study by Levesque et al) or had a screening algorithm using serum GM (Prattes et al) which is less sensitivity as discussed in the diagnostic considerations section (6.1). We added a sentence to make this clear.

Line 245; What does RRT mean?

            Response: Renal replacement therapy. The first use for this was on line 196 “renal replacement therapy (RRT)”

Section 4.4. The literature review in this section seems not to be based on recent literature; you cite several papers from 2010 and 2012, which is fine, but I would recommend confronting the data from those references with recent papers. Such a comparison of data will definitely strengthen your paper.

Response: Thank you for this feedback. We reviewed the pubmed results for “invasive aspergillosis” and “critical illness” or “COPD” or “ICU” or “COVID” in the last 5 years, and the relevant results were incorporated into this review where possible.

Lines 436-440, the role of beta-D-glucan should be elaborated. What are the pros and cons of this marker? Why is it not commonly used in clinical practice like GM?

Response: The section was expanded to include this information, thank you for this feedback

Reviewer 2 Report

The topic is very interesting; the information is presented clearly and is based primarily on articles published in the last 25 years. The title is appropriate for the manuscript's content, and the information provides a broad and descriptive overview of the topic. However, I note some aspects that could be improved:

  1. The abstract is incomplete, as it does not present the results or conclusions of the literature review.
  2. In the Introduction, lines 26-27, please expand on the information about the etiology of invasive aspergillosis.
  3. I think the review is good, but since it is not a systematic review, it has some limitations that the authors should point out in the manuscript, for example, information bias.
  1. Figures 1 and 2 should include a figure caption with a more comprehensive and descriptive description. In Figure 3, place the meaning of the acronyms or abbreviations in the figure caption.
  2. Citations and references should comply with the journal's guidelines for authors.
  3. Scientific names should be italicized.

Author Response

Reviewer 2

Major comments

The topic is very interesting; the information is presented clearly and is based primarily on articles published in the last 25 years. The title is appropriate for the manuscript’s content, and the information provides a broad and descriptive overview of the topic. However, I note some aspects that could be improved:

Response: Thank you for this positive feedback

The abstract is incomplete, as it does not present the results or conclusions of the literature review.

Response: Thank you for this feedback. The abstract was expanded to include information about the results and conclusions of the review

In the Introduction, lines 26-27, please expand on the information about the etiology of invasive aspergillosis.

Response: This is covered in detail in the following section (2. Pathogenesis of invasive aspergillosis) and including it here would be repetitive.

I think the review is good, but since it is not a systematic review, it has some limitations that the authors should point out in the manuscript, for example, information bias.

Response: Thank you for this feedback. A limitations section was added towards the end of the review to highlight this.

Detailed comments

Figures 1 and 2 should include a figure caption with a more comprehensive and descriptive description. In Figure 3, place the meaning of the acronyms or abbreviations in the figure caption.

Response: Thank you for this feedback, this was done.

Citations and references should comply with the journal's guidelines for authors.

Scientific names should be italicized.

Response: Thank you for this feedback. The references were formatted according to MDPI guidelines (https://www.mdpi.com/authors/references). If any issues, we hope the editorial team can work with us on fixing these issues once the article is accepted for publication

Reviewer 3 Report

The manuscript does not bring, in my opinion, any new information to the already existing literature. I have written more detailed comments below. While the article could be improved and resubmitted in the future, I do not believe it is suitable for publication in its current form.

I think the biggest weakness of this review manuscript is that it doesn't bring any new information to the currently existing literature, or, since it is in fact a review, it does not sumarize or point out anything new, that is not already known by specialists from the field (and even just general practitioners that find this infection in their current practice). 

The manuscript might have been more suitable if it was, at least, a systematic review. But it is simply a story of the findings the authors selected from literature about risk factors for developing aspergillosis. Even more, I do believe that the information is not detailed enough, just briefly going into general, widely studied things. For example in the pathogenesis chapter, the exact mechanisms are just briefly mentioned, not described or classified in any way, and just superficially addressed. In chapter 3, I do not believe that a whole table was needed for something that is simply an enumeration and could have easily been a list. Even more, the authors could have at least mentioned in the table the articles where they took the information regarding each risk factors from or add something that would have made the information important enough to be added to a table. Basically, again, it is just an enumeration of generally known information that you can already find with a quick search on the internet.

There are places in the article where references are not added (e.g., lines 102-113).

Regarding the emerging risk factors, which as I understand are the main novelty brought by the article, I believe they are not new at all. COVID-19 and Influenza have both been known risk factors for acquiring pulmonary aspergillosis for quite some time, as a lot of studies regarding this matter were happening during the pandemic. Same thing for critical illnesses. There is nothing new there. It is a well-known fact that people develop mold infections when they have several multiple comorbidities and their immune system is affected. Also, respiratory affections (like COPD), immunomodulatory medicine and solid malignancies are also common known risk factors for aspergillosis. 

In the diagnostic part of the review, nothing is mentioned regarding the differentiation between infection and colonization, which is actually the biggest challenge in diagnosing asperigillosis.

Author Response

Reviewer 3

Major comments

The manuscript does not bring, in my opinion, any new information to the already existing literature. I have written more detailed comments below. While the article could be improved and resubmitted in the future, I do not believe it is suitable for publication in its current form.

Response: Thank you for taking the time to review our manuscript. Please see our detailed response to this point below.

Detailed comments

I think the biggest weakness of this review manuscript is that it doesn't bring any new information to the currently existing literature, or, since it is in fact a review, it does not sumarize or point out anything new, that is not already known by specialists from the field (and even just general practitioners that find this infection in their current practice).

Response: This review is a comprehensive overview of the emerging risk factors for invasive aspergillosis and is geared towards providers working both in and outside of the intensive care unit. It raises awareness of a condition that is likely frequently missed, as clinicians may not be aware of all the (frequently co-existing) risk factors and may not consider testing or treating for IPA as a result. Feys et al [1] likened it to attempting to diagnose a myocardial infarction but with limited use of troponins and EKGs, an analogy that we strongly agree with. Lack of awareness of the emerging risk factors is likely a major reason for missed diagnosis and the low utilization of diagnostic studies including bronchoscopy in these patients. IPA was diagnosed premortem in only 27% of fatal cases in an autopsy study from 2022 [2]. IPA still shows up as a missed diagnosis on autopsies. With the increasing significance of viral pneumonias and the increasing complexity of patients admitted to the ICU, this is likely to become a bigger issue in the future. While it is not a groundbreaking study, it is a review of a complex topic and achieves its stated aims.

References:

  1. Feys, S., Hoenigl, M., Gangneux, J. P., Verweij, P. E., & Wauters, J. (2024). Fungal Fog in Viral Storms: Necessity for Rigor in Aspergillosis Diagnosis and Research. American journal of respiratory and critical care medicine, 209(6), 631–633.
  2. Mudrakola, H. V., Tandon, Y. K., DeMartino, E., Tosh, P. K., Yi, E. S., & Ryu, J. H. (2022). Autopsy study of fatal invasive pulmonary aspergillosis: Often undiagnosed premortem. Respiratory medicine, 199, 106882.

The manuscript might have been more suitable if it was, at least, a systematic review. But it is simply a story of the findings the authors selected from literature about risk factors for developing aspergillosis. Even more, I do believe that the information is not detailed enough, just briefly going into general, widely studied things. For example in the pathogenesis chapter, the exact mechanisms are just briefly mentioned, not described or classified in any way, and just superficially addressed.

Response: Systematic reviews are increasingly needed to summarize the available literature on the topic but they are often better suited to answer a specific question eg what is the accuracy of serum GM in the diagnosis of IPA. Attempting a systematic review summarizing this broad topic was not feasible within the time constraints faced by our team. Our goal was to provide a comprehensive overview to raise awareness of the non-classic risk factors for IPA, geared towards practicing providers and learners working in or outside the ICU who may not be very familiar with the topic. We believe it achieves this aim well. The reviewer may be an expert in the field and thus does not see the benefit of this review to most general practitioners. Lastly, going into further details on the pathogenesis, for example, is beyond our scope, as it is not clinically relevant to the target audience.

In chapter 3, I do not believe that a whole table was needed for something that is simply an enumeration and could have easily been a list. Even more, the authors could have at least mentioned in the table the articles where they took the information regarding each risk factors from or add something that would have made the information important enough to be added to a table. Basically, again, it is just an enumeration of generally known information that you can already find with a quick search on the internet.

Response: Chapter 3 was intended to briefly mention the classic risk factors of IPA, before going into the main part of the review. We did not want to add numerous more references to briefly mention the well-known classic risk factors for IPA. We can remove this table and instead mention these risk factors in the text of chapter 3. We provided a references (34-37) for readers who want to delve into this topic.

There are places in the article where references are not added (e.g., lines 102-113).

Response: Thank you, this was corrected. We apologize for the oversight.

Regarding the emerging risk factors, which as I understand are the main novelty brought by the article, I believe they are not new at all. COVID-19 and Influenza have both been known risk factors for acquiring pulmonary aspergillosis for quite some time, as a lot of studies regarding this matter were happening during the pandemic. Same thing for critical illnesses. There is nothing new there. It is a well-known fact that people develop mold infections when they have several multiple comorbidities and their immune system is affected. Also, respiratory affections (like COPD), immunomodulatory medicine and solid malignancies are also common known risk factors for aspergillosis.

Response: Thank you for your feedback. Yes, this literature review is a review of known information and is written to provide readers with an overview of this topic. These risk factors frequently coexist in patients, yet unfortunately, there continues to be a gap in clinical practice with many clinicians not routinely testing for aspergillosis or considering bronchoscopy in patients who meet criteria outlined by the Invasive Fungal Diseases in Adult Patients in Intensive Care Unit (FUNDICU) 2024 consensus.

In the diagnostic part of the review, nothing is mentioned regarding the differentiation between infection and colonization, which is actually the biggest challenge in diagnosing asperigillosis.

Response: Thank you for this feedback. While the diagnosis section did not specifically mention the difficulty in differenting between infection and colonization, the review discusses that it is difficult to ascertain as the only way to diagnose definite/proven IPA is by histopathologic, cytopathologic, or direct microscopic examination showing fungal invasion (the first paragraph of the section, lines 412-422). Given that this is not possible in most patients, we describe in detail the criteria to diagnose probable IPA. However, we added a section to specifically discuss this.

Round 2

Reviewer 2 Report

The authors successfully made the suggested modifications.

The authors successfully made the suggested modifications.

Author Response

Thank you for taking the time to review our manuscript. 

Reviewer 3 Report

I would like to thank the authors for the point-by-point reply to my previous review. Even though the answers are valid, I still do not agree that this manuscript brings anything new to the already existing literature, therefore, I do not think it should be published. 

I have provided a detailed answer below.

Author's response #1: This review is a comprehensive overview of the emerging risk factors for invasive aspergillosis and is geared towards providers working both in and outside of the intensive care unit. It raises awareness of a condition that is likely frequently missed, as clinicians may not be aware of all the (frequently co-existing) risk factors and may not consider testing or treating for IPA as a result. Feys et al [1] likened it to attempting to diagnose a myocardial infarction but with limited use of troponins and EKGs, an analogy that we strongly agree with. Lack of awareness of the emerging risk factors is likely a major reason for missed diagnosis and the low utilization of diagnostic studies including bronchoscopy in these patients. IPA was diagnosed premortem in only 27% of fatal cases in an autopsy study from 2022 [2]. IPA still shows up as a missed diagnosis on autopsies. With the increasing significance of viral pneumonias and the increasing complexity of patients admitted to the ICU, this is likely to become a bigger issue in the future. While it is not a groundbreaking study, it is a review of a complex topic and achieves its stated aims.

Response: Thank you for your response. While I do understand your statement, and the fact that this review is written for people working in the Intensive care Unit (and outside), I am still not completely convinced on the relevancy of the chosen topic for a review. First of all, Aspergillus is one of the most common molds. According to Viegas et al. ("Aspergillus spp. prevalence in Primary Health Care Centres: Assessment by a novel multi-approach sampling protocol"), Aspergillus genus is responsible for over 80% of pulmonary invasive fungal infections in humans. Therefore, it is quite a frequent and widely studied fungal infection. However, since your reasoning is that this comprehensive review is addressed to ICU personnel who might not be aware of this condition, please find below a few articles researching this same topic published in recent literature:

https://link.springer.com/article/10.1186/s13613-021-00923-4

https://www.frontiersin.org/journals/medicine/articles/10.3389/fmed.2021.753659/full

https://pmc.ncbi.nlm.nih.gov/articles/PMC8441237/

Please also refer to this article - https://www.mdpi.com/2309-608X/11/1/70 published on 17 January 2025 in the same journal that also broadly describes the same topic. I am sorry, but I do not agree with the authors and I still don't consider that this review brings anything new to the field.

Author's response #2: Systematic reviews are increasingly needed to summarize the available literature on the topic but they are often better suited to answer a specific question eg what is the accuracy of serum GM in the diagnosis of IPA. Attempting a systematic review summarizing this broad topic was not feasible within the time constraints faced by our team. Our goal was to provide a comprehensive overview to raise awareness of the non-classic risk factors for IPA, geared towards practicing providers and learners working in or outside the ICU who may not be very familiar with the topic. We believe it achieves this aim well. The reviewer may be an expert in the field and thus does not see the benefit of this review to most general practitioners. Lastly, going into further details on the pathogenesis, for example, is beyond our scope, as it is not clinically relevant to the target audience.

Response: Thank you for your reply. Firstly, I unfortunately believe that a time constraint is not a valid excuse when trying to publish a scientific article. I have checked and there is no funding for this article, therefore I do not see a reason why the authors would have a time limit or why this should generally affect the quality of a good scientific article. Secondly, if the topic was too broad, the authors should have chosen a more focused area or a more specific part of this broad topic. I understand, again, what is the aim of your article, but this journal focuses on all kinds of emerging fungal infections, pathogenesis and experimental studies and is mostly addressed to experts in the field of mycology or clinicians interested in diagnosing fungal infections. Also, the authors mention that they do not want to go into more detail regarding pathogenesis, as it is not the main topic of the article, but they submitted the article to a special issue entitled Fungal Pathogenesis and Disease Control.

Author's response #3: Chapter 3 was intended to briefly mention the classic risk factors of IPA, before going into the main part of the review. We did not want to add numerous more references to briefly mention the well-known classic risk factors for IPA. We can remove this table and instead mention these risk factors in the text of chapter 3. We provided a references (34-37) for readers who want to delve into this topic.

Response: Thank you for addressing this comment. As mentioned by the authors in the reply to the first round of review, the main purpose of a review is "to summarize the available literature on the topic". Therefore, while doing the research, the main aim would be to read as much as possible on a certain topic, so saying that you did not want to add too many references contrasts with the main purpose of the manuscript, which is to summarize as much literature on your set topic as possible. 

Author Response

Response to reviewer 3

Once again, thank you for taking the time to review our manuscript and responses.

Reviewer comment 1:

“Thank you for your response. While I do understand your statement, and the fact that this review is written for people working in the Intensive care Unit (and outside), I am still not completely convinced on the relevancy of the chosen topic for a review. First of all, Aspergillus is one of the most common molds. According to Viegas et al. ("Aspergillus spp. prevalence in Primary Health Care Centres: Assessment by a novel multi-approach sampling protocol"), Aspergillus genus is responsible for over 80% of pulmonary invasive fungal infections in humans. Therefore, it is quite a frequent and widely studied fungal infection. However, since your reasoning is that this comprehensive review is addressed to ICU personnel who might not be aware of this condition, please find below a few articles researching this same topic published in recent literature:

https://link.springer.com/article/10.1186/s13613-021-00923-4

https://www.frontiersin.org/journals/medicine/articles/10.3389/fmed.2021.753659/full

https://pmc.ncbi.nlm.nih.gov/articles/PMC8441237/

Please also refer to this article - https://www.mdpi.com/2309-608X/11/1/70 published on 17 January 2025 in the same journal that also broadly describes the same topic. I am sorry, but I do not agree with the authors and I still don't consider that this review brings anything new to the field.”

Response

While many studies have been published on invasive aspergillosis, we do not believe there is a recent comprehensive review that encompasses all emerging risk factors. We thank you for giving us specific examples of recently published literature, but here is how they are different from this review: 

  1. Montrucchio et al (2021) is a review article that specifically (and only) discusses IPA due to severe COVID-19 infection.
  2. Xu et al (2021) is a retrospective cohort study (not even a review) of IPA in severe COVID-19
  3. The third link is actually the same article as the first link
  4. Vazuquez et al (2025) only discusses the ICU as a risk factor

Our article is a comprehensive overview of all the emerging risk factors, including critical illness, viral pneumonia, COPD, cirrhosis, new anticancer therapies such as CAR-T cell therapies. We highlight the fact these conditions can overlap in patients.

As per the editor's recommendation, we added a paragraph in the introduction and conclusion sections to specifically highlight the additive value of this review article to the currently published literature.

Reviewer comment 2

“Thank you for your reply. Firstly, I unfortunately believe that a time constraint is not a valid excuse when trying to publish a scientific article. I have checked and there is no funding for this article, therefore I do not see a reason why the authors would have a time limit or why this should generally affect the quality of a good scientific article. Secondly, if the topic was too broad, the authors should have chosen a more focused area or a more specific part of this broad topic. I understand, again, what is the aim of your article, but this journal focuses on all kinds of emerging fungal infections, pathogenesis and experimental studies and is mostly addressed to experts in the field of mycology or clinicians interested in diagnosing fungal infections. Also, the authors mention that they do not want to go into more detail regarding pathogenesis, as it is not the main topic of the article, but they submitted the article to a special issue entitled Fungal Pathogenesis and Disease Control.”

Response

A literature review is useful for summarizing the current state of knowledge for this broad topic. A systematic review would indeed be better IF our objective was to answer a specific question. Our objective was clear: to summarize and update providers about the emerging risk factors for a serious yet frequently missed condition. This review achieves this. A systematic review would not have necessarily done this better. However, literature reviews have their limitations, which is highlighted in the limitations section included in the manuscript. 

Additionally, we would like to clarify that this article was not specifically sent to the special issue entitled Fungal Pathogenesis and Disease Control

Reviewer comment 3

“Thank you for addressing this comment. As mentioned by the authors in the reply to the first round of review, the main purpose of a review is "to summarize the available literature on the topic". Therefore, while doing the research, the main aim would be to read as much as possible on a certain topic, so saying that you did not want to add too many references contrasts with the main purpose of the manuscript, which is to summarize as much literature on your set topic as possible.”

Response 

The reviewer may have misunderstood the prior response. The purpose of the manuscript is to discuss the emerging risk factors of IPA, as well as the pathophysiology, incidence, outcomes, diagnostic and treatment considerations. Chapter 3 was intended to briefly mention the classic risk factors of IPA. It is not our intention to write a review about the classic risk factors. We provided references (34-37) of review articles on classic risk factors for readers who want to delve into this part of the topic.
